# Connexins and Pannexins—Similarities and Differences According to the FOD-M Model

**DOI:** 10.3390/biomedicines10071504

**Published:** 2022-06-25

**Authors:** Irena Roterman, Katarzyna Stapor, Piotr Fabian, Leszek Konieczny

**Affiliations:** 1Department of Bioinformatics and Telemedicine, Jagiellonian University—Medical College, Medyczna 7, 30-688 Kraków, Poland; 2Department of Applied Informatics, Faculty of Automatic, Electronics and Computer Science, Silesian University of Technology, Akademicka 16, 44-100 Gliwice, Poland; katarzyna.stapor@polsl.pl; 3Department of Algorithmics and Software, Faculty of Automatic, Electronics and Computer Science, Silesian University of Technology, Akademicka 16, 44-100 Gliwice, Poland; piotr.fabian@polsl.pl; 4Chair of Medical Biochemistry—Jagiellonian University—Medical College, Kopernika 7, 31-034 Kraków, Poland; mbkoniec@cyf-kr.edu.pl

**Keywords:** connexin, pannexin, membrane proteins, transmembrane channel, hydrophobicity

## Abstract

Connexins and pannexins are the transmembrane proteins of highly distinguished biological activity in the form of transport of molecules and electrical signals. Their common role is to connect the external environment with the cytoplasm of the cell, while connexin is also able to link two cells together allowing the transport from one to another. The analysis presented here aims to identify the similarities and differences between connexin and pannexin. As a comparative criterion, the hydrophobicity distribution in the structure of the discussed proteins was used. The comparative analysis is carried out with the use of a mathematical model, the FOD-M model (fuzzy oil drop model in its Modified version) expressing the specificity of the membrane’s external field, which in the case of the discussed proteins is significantly different from the external field for globular proteins in the polar environment of water. The characteristics of the external force field influence the structure of protein allowing the activity in a different environment.

## 1. Introduction

The specific group of proteins that act in the membrane surrounding linking the external environment with the cytoplasm is called the transmembrane proteins [1]. Many transmembrane proteins function as gateways to permit the transport of specific substances across the membrane. The transport of molecules through the membrane can take two forms: the use of a channel passing the selected molecules and ions, and the pump transporting ions against the gradient of concentration [2,3,4,5,6]. The second type frequently requires significant conformational changes, ensuring the process called “active transport” [7].

Membrane proteins require the exposure of hydrophobic residues on their surfaces to adapt to the hydrophobic environment. In contrast to this, in water-soluble proteins, the surface is covered with the polar residues while the hydrophobic ones are concentrated in the central part, forming a hydrophobic core [8]. There are several key factors contributing to the difference in hydrophobicity distribution observed between membrane proteins and soluble proteins: (1) the polar aquatic environment and hydrophobic membrane environment and (2) the folding process which is entropy-driven. Thus, hydrophobicity distribution ensures the solubility of proteins operating in the aqueous medium. Such a structure is also the effect of the active participation of the aquatic environment in the folding process. The presence of polar water directs the hydrophobic residues towards the central part of the protein. Polar amino acids are preserved on the surface, creating an entropy-favorable system at the protein-water interface. In transmembrane proteins, apart from the exposure of hydrophobic residues on the surface, the presence of free space in the central part of the molecule—the channel—shows a low hydrophobicity, in contrast to the water-soluble proteins [9]. The stabilization of membrane proteins in a hydrophobic environment creates numerous research problems due to the need for the presence of their natural environment. Nevertheless, the structures of membrane proteins appear in the PDB database [10] thanks to the use of the CryoEM technique [11] as well as X-ray [12,13,14,15,16]. Stabilization is ensured by the presence of, for example, detergent micelles in contact with the analyzed protein or in a complex with other lipid-like molecules [17,18,19]. Currently, lipids in the form of nanodiscs and liposomes can be used to stabilize membrane proteins for both structural and functional analysis [20,21,22,23]. The progress in the development of experimental techniques influences and makes possible the application of theoretical investigations, including the simulation of molecular dynamics [24,25,26,27,28,29,30,31,32]. Accessibility to a permanently grooving database of membrane proteins makes this research easier and more fruitful [33]. The large size of the membrane protein databases allows the investigation even of the systems biology scale [34,35,36]. Different techniques are applied to solve the main problem of the necessary presence of a hydrophobic environment [37,38,39]. The current progress in the research on membrane protein structures is presented in some excellent reviews [40].

Connexins and pannexins as well as innexins playing the common role of channels represent the special position among the membrane proteins [40].

Connexins or gap junction proteins constructed in form of six monomers are transmembrane proteins that assemble to form vertebrate gap junctions. Each gap junction is composed of two hemichannels, which are located in the plasma membrane of adjacent cells. Pannexins are channel membrane proteins formed by six or eight monomers. The way connexins and pannexins form structures within the membrane are different from each other. Connexins, unlike pannexins, have the ability to transport from the inside of one cell to the inside of the other, creating a channel running through the connected membranes of two closely approached cells. It is possible thanks to the ability to form two hemichannels linked together. Oligomeric structures of pannexins anchored in a single plasma membrane enable the construction of a channel between the cytosol and extracellular space. [41]

For the analysis of the proteins discussed here, a model called fuzzy oil drop (FOD) was used which treats the hydrophobicity distribution in a protein as a result of the influence of the environment in which the protein has been folded or in which it operates. This model was originally proposed solely for the structuring of proteins in the aquatic environment. Its modified version (FOD-M) takes into account the presence of factors other than polar water, such as the presence of hydrophobic factors in the cell membrane in particular. The correctness of the operation of the FOD-M model was demonstrated in the analysis of other transmembrane proteins of the all-helical type—rhodopsin [42], Beta-barrel—porin [43], mechanosensitive channels [44]. The assessment of the hydrophobicity distribution, and especially the measure of the degree of deviation from the idealized distribution observed in the globular proteins with a regular hydrophobic core, allows for determining the degree of participation of non-water factors in the structuring of membrane proteins.

The object of analysis in the present work is the representative of connexins—human connexin Gap junction beta-2 (PDB ID 6L3T) [45]—and human pannexin-1 (PDB ID 7F8J [46]—CryoEM techniques were applied in both cases.

The selected Cx31.3/GJC3 to represent the connexins is the exceptional one (PDB ID 6L3T) [45] due to the inability to generate the intercellular channel. It means the selected example of connexins appears solely in form of hemichannel. This form of connexin makes the comparative analysis with the pannexin reliable just limited to the search for structural similarity with respect to the structuralization in a specific environment. It delivers the possibility for other similarities/differences searches comparing the ability to generate the intercellular gap junction. It means the similarity search of the extra-cellular part of these proteins becomes an interesting object of independent analysis, which is planned to be continued.

## 2. Results

It is important for comparative analysis of objects to represent possibly comparable status including the same species and the same research technique in particular. The connexin and pannexin representatives discussed here are of human origin. The research technique for identifying the structure is CryoEM. The selected connexin structure is available in PDB in two versions: in the presence of Ca^2+^ ions and in the absence of these ions, the presence of which is critical for the activity of this protein. When comparing the connexin’s structure with pannexin’s one, the question arises about the form that should be compared. Before such analysis was performed, the influence of the presence of Ca^2+^ ions on the connexin’s structure was identified.

### 2.1. The Influence of Ca^2+^ Ioins on Connexin’s Structure

The values of RD and K parameters for the form with Ca2 + ions show higher values of RD and K (Ca^2+^-absent: RD = 0.743 and K = 1.4 while Ca^2+^-present results in values of RD = 0.767 and K = 1.7) (Figure 1). The presence of calcium ions is required for the activity of the protein and makes it different from pannexin, the structure of which has been obtained without additional external factors. Therefore, it was decided to choose a Ca^2+^-free structure for the comparative analysis with the pannexin’s structure. To illustrate the effect of the presence of Ca^2+^, the set of M profiles for both forms of this protein is shown (Figure 1).

The summary of the M profiles (Figure 1) reveals the relaxation of the center (channel location)—two N-terminal peaks with a reduced level of hydrophobicity concentration. The two local C-terminal maxima remain unchanged, suggesting that the presence of Ca^2+^ did not affect the structuring of these chain sections. The list of profiles (Figure 1) can be used to assess the impact of an additional environmental factor, which makes the analysis of M profiles universal.

### 2.2. The General Characteristics of Connexin’s Hemichannels and Pannexin’s Channels

By analyzing the localization of both discussed transmembrane proteins, many similarities can be identified, including the exposure of analogous parts of the complexes to the extra-membrane zone (Figure 2A). Another similarity is the construction based on the all-helical part embedded in the membrane. The extra-membrane part directed outside the cell differs in some respects. This part includes a β-sheet made of three β-strands. The difference is in the number of chains in the complex: 7 in the pannexin which comprises the positions 2–356 (with the section 159–193 unresolved) and 6 in connexin (chain 2–221 with the section 108–126 unresolved) (Figure 2). The missing fragments belong to the external part of the chains directed toward the cellular area. Common to both compared proteins is the presence of symmetry. Chains are arranged according to the 7-fold axis in pannexin and 6-fold one in connexin, which is the simple consequence of the chain number present in the homo-complex (Figure 2C,D). For obvious reasons, the size of the complex appears smaller for connexin with respect to pannexin. 

The visualization given in Figure 2A,B identifies the membrane helices (magenta) in both complexes as well as the part of the complex directed outside the membrane with 3-strands β-sheet in each chain (yellow in Figure 2A,B). The part of the complex exposed outside the cell appears bigger in pannexin in comparison with connexin. The packing of chains appears slightly different in compared complexes. The problem of channel size—different as shown in Figure 2C,D shall be the object of independent analysis. The presentation visualizing the hydrophobicity level (option available in the PDB database) shows the specificity of discussed molecules characterized by the exposure of hydrophobicity on the surface (Figure 2E,F)

The multiple sequence alignment of the two amino acid sequences (6L3T, 7F8J) using the BLASTP program [47,48] showed no sequence similarity.

Since the object of analysis in this paper is the identification of differences with respect to the role of the hydrophobic environment, the analysis carried out on the basis of the fuzzy oil drop model (FOD) [49] and its modified version FOD-M [42,43,44] provides an assessment of the degree of adjustment of the hydrophobicity distribution in relation to the idealized one, which is the distribution consistent with the 3D Gauss function. For obvious reasons, the FOD-M model was used to describe the protein structures discussed here. The values of the parameter K measuring the participation of the factors different from the polar water were determined. In this particular case, it is the presence and influence of the hydrophobic environment of the membrane.

The summary in Table 1 presents the status of the complexes (hemichannels) and of single chains treated as individual structural units. The assessment of the complexes is based on the generation of a 3D Gauss function spread over the entire complex. The single-chain status is defined as a 3D Gaussian function spanning a single chain.

Table 1 shows the calculated values of the RD and K parameters in the FOD-M model for the entire structural units—complexes and the single chains as well as the segments defined by disulfide bonds (here, the hemichannel for connexin and channel for pannexin is taken as the structural unit). As expected, the values of parameters RD and K for both complexes show very high values. This is due to the specific nature of the environment, which significantly differs from the polar aquatic environment. The low level of hydrophobicity in the central part at the site of the expected hydrophobic core and the exposure of hydrophobic residues are the reasons for the high values of RD and K parameters for the analyzed complexes. This situation is illustrated by the T, O, and M distribution profiles for the given K value shown in Figure 3. The M profile demonstrates the T distribution—the idealized one—after modification introduced by the inverse of the 3D Gauss function, which expresses the presence of a hydrophobic environment. The M profile is the effect of the hydrophobic environment directing hydrophobic residues toward the surface. The status of the residues involved in the interactions with molecules is also given in Table 1. In the case of pannexin with molecule **LBN**—1-Palmitoyl-2-Oleoyl-Sn-Glycero-3-Phosphocholine [(2r)-2-[(9z)-9-Octadecenoyloxy]-3-(Palmitoyloxy)propyl 2-(Trimethylammonio) ethyl phosphate] having formula C_42_H_82_NO_8_P (3 per change on average) and in connexin—Lauryl maltose neopentyl glycol [2,2-Didecylpropane-1,3-Bis-B-D-Maltopyranoside] having the formula: C_47_H_88_O_22_ (1 per change on average).

Despite the lower number of ligands in connexin, the status of interacting residues with them is expressed by a higher RD value, which means a higher influence on the status related to the external environment.

An analysis of the profiles representing complexes of connexin and pannexin reveals the similarities of profiles representing the typical status for membrane proteins. The higher hydrophobicity level on the surface—due to hydrophobic membrane surrounding, significant deficiency of hydrophobicity in positions of expected hydrophobic core—due to the presence of channel in the central part of the complex is observed in all membrane proteins [42,43,44].

The role of the N-terminal half seems to be similar as responsible for channel construction– high hydrophobicity deficiency in connexin as well as in pannexin. The expected high level in the central part (high T values) of the complex is not met by real impacts revealing a clear hydrophobicity deficit. The C-terminal half of the chains represents the status of relatively high accordance between T and O, which suggests the role of stabilization in this area; however, the higher-than-expected hydrophobicity suggests the interaction with the membrane environment.

The profile for the M distribution takes the form close to a line parallel to the horizontal axis. This is a consequence of the high K value expressing the necessary modification of the environment with respect to the water surrounding.

Summarizing, it can be concluded that the T, O, and M distributions represent a typical form for membrane proteins serving as a channel. A high K value indicates a significant deviation from the distribution typical of water-soluble proteins. The segments expected to be components of the hydrophobic nucleus represent a significant deficit in hydrophobicity (channel free space). There is also an excess of hydrophobicity in the surface sections, which results from the need for stabilizing hydrophobic interaction with the surrounding membrane.

The status of a single chain treated as an individual structural unit (3D Gauss function spanned on a single chain) is expressed by the T, O, and M profiles with the given values of the K parameter (Figure 4).

The comparison of profiles representing the status of individual chains suggests the role of particular chain fragments in the folding process. The N-terminal and C-terminal local maxima in connexin represent the status of O similar to T. It suggests that the folding process for these fragments followed the water environment scenario. Two central-local maxima, however, represent the deficiency of hydrophobicity.

Comparing this set of profiles with the ones shown for the complex one may cause speculation that the participation in channel construction in the complex was already prepared in the monomer structure of this chain. The participation in channel construction of the fragment representing the N-terminal local maxima appeared after complex construction. A similar interpretation can be given for the fragment representing the third local maxima.

Characteristics of individual chain in pannexin identifies the N-terminal half of the chain (two N-terminal maxima) as channel constructors with evident hydrophobicity deficiency. The C-terminal half appears as representing rather high O and T similarity however with a higher hydrophobicity level on the surface. It seems that this part of chains is responsible for membrane interaction. The comparison between individual chain status and the one in complex in pannexin suggests the presence of preparation for channel construction with a high deficiency in the N-terminal half (1–100 amino acid positions). The C-terminal part of the chain represents a similar status; however, the high level of hydrophobicity suggests the preparation for interaction with the membrane.

The status of the segments defined by the positions of the disulfide bonds (Table 1) reveals their role as a stabilizer of the structure far from the globular protein system, giving these segments a structuring prepared to perform a biological function in the final structure within the complexes. The single-chain status of these fragments supports this observation. Similarly, the residues involved in P-P (protein-protein) interactions within individual chains represent a status typical to the exposed residues for the interaction with other chains introducing local maladjustment to the expected distribution. The reverse is the case in the complex where the status of the residues involved in the P-P interaction shows a lower RD value compared to the status of the rest of the molecule (No P-P). The status of individual chains in pannexin expresses the high exposure of hydrophobicity on the surface (RD for P-P 0.821 with RD for No P-P = 0.652) suggests that the large part of the surface with exposed hydrophobicity is “consumed” by inter-chain interaction to construct the inter-chain interface. The high value for No P-P part status expresses the exposure of hydrophobicity oriented on the interaction with the membrane as it is observed in other membrane proteins [42,43,44]. This proportion is lower in connexin.

The fragments with an excess of hydrophobicity in positions engaged in inter-chain interaction visualize the aim of this status observed already in the individual chain. These fragments are prepared for complexation. The fragments representing the hydrophobicity excess without engagement in protein-protein interaction are responsible for stabilization in the membrane surrounding. The analysis of the status of the individual chain makes possible the preparation of the initial structure to play a further role in the form of aim-oriented interaction.

The status of residues interacting with ligands (P-Ligand and No Ligand)—in this case, they are the molecules mimicking the membrane that appears comparable in both compared proteins. The higher value for No-Ligand means that the disorder related to the channel formation is the stronger one that was caused by exposure to hydrophobic residues on the surface. This conclusion can, however, not be treated as general due to the limited number of membrane-mimicking molecules.

### 2.3. The Comparative Analysis of Connexin’s Hemichannels and Pannexin’s Channels

A comparative analysis was performed in this work using the hydrophobicity distributions both in a single chain and in a complex. This analysis was performed in a different way from that presented above. The set of T-distributions for two individual chains and the set of *T*-distributions for the complexes were analyzed. The *T*-distributions are a reference revealing the expected status resulting from the location within the analyzed body. In the case of the currently compared chains from the *T* distributions, it is possible to obtain information on the expected involvement of the relevant sections in the formation of the core and identify sections located on the surface. By means of such sets, it is possible to identify the segments which are expected to fulfill similar structural roles.

The comparison of the *O* distributions for the two compared chains allows us to determine to what extent—comparable or different—a given goal has been or has not been achieved. It is also possible to quantify the degree of deviation from the expected, idealized status.

#### 2.3.1. Status of Individual Chain

In the set of *T* profiles for a single chain, 4 sections with analogous status were identified. This means that theoretically, the identified segments should play similar roles in the structure of the discussed chains. Analogous segments determined on the basis of T profiles are given in Table 2. (Figure 5).

Assuming a globular structure with a centric hydrophobic core, five segments with a profile corresponding to this status are expected. To what extent the structural role of these analogous stretches is fulfilled, is shown by the profiles in Figure 5.

The compared profiles reveal differences in the way the O distributions adjust/maladjust against the T reference distributions in the fragments designated as analogous (Figure 5). Their 3D presentation is shown in Figure 6. Visualizes the location in the structure which appear similar in compared chains despite the different length of chains.

Comparative analysis of the single-chain status reveals differences in the evaluation and role of individual segments. The predicted location of the stabilizing hydrophobic core concerns different segments with a different location in the protein structure (Figure 6). On the other hand, the degree of task accomplishment (formation of the hydrophobic core) is for obvious reasons not realized, but in a different way. The status of the segments expected as components of the core in pannexin shows a significant deviation from the idealized structure, other than in the case of connexin (fragments 2–21). The status of the sections (22–68)/(61–107) and (96–146)/(127–177) in pannexin/connexin are also different. Here the higher values of RD and K show connexin, where these sections are directly involved in the construction of the canal. The status of the sections (202–245)/(178–221), however, is comparable. These segments having relatively lower values of RD are therefore part of the stabilization of the structure as such. In contrast, all other segments in connexin show a significant mismatch, which can be interpreted as a function-related role for this part of the chain (anchoring in the womb and presence of a channel).

The difference in the assessment of the status of analogous fragments in comparable proteins is also due to differences in their length. A single pannexin chain consists of 321 aa, whereas a connexin chain is composed of 203 aa (not counting the missing segments not visible in the CryoEM technique).

Therefore, in the connexin chain, all roles (hydrophobic surface and channel presence) are performed by the analyzed segments. On the other hand, in pannexin, the analyzed segments are components of molecular stability fulfilling the conditions of low *RD* and low K values.

#### 2.3.2. Status of Complex

The comparison of T profiles for the two complexes (represented in Figure 7 by chain A since all others are described by the identical profiles) reveals the expectation for segments that can be treated as analogous.

The status of fragments 2–31 in pannexin appears surprisingly accordant with idealized distribution (RD < 0.5) as well as the fragment (99–148) in pannexin. The most diverse status in the complex is represented by the analogous sections (99–148)/(127–176) in pannexin/connexin, whereas in pannexin the status is closer to the one ordered according to the FOD model.

The question arises, however, to what extent these objectives imposed by the idealized distribution are met by the corresponding sections. This can be read from the *O* profiles for the two compared chains (complexes). A significant separation of actual status from expected status becomes apparent. From the analysis given above a significant reduction in the hydrophobicity level at the sites identified as a potential hydrophobic core is apparent. Similarly, at sites expected to be surface polar, the level of hydrophobicity is comparable to that in the protein interior. 

The status expressed by RD and K values determines the extent to which a given target was or was not achieved in a given section.

A summary of the results given in Table 2 reveals differences between the status of analogous sections in the compared proteins (Figure 7 and Figure 8).

Figure 8 reveals a similar role in hydrophobicity distribution in a particular chain in context with a complex structure. The expected high concentration of hydrophobicity informs about the central location of a particular fragment. The selected fragments seem to play similar roles in both constructions; however, this role is achieved by the opposite orientation of chains. The color arrows in Figure 7 visualize this relation. The different orientations of individual chain fragments may suggest a different course of the protein folding process in the membrane environment. It probably depends on the orientation during chain introduction into the membrane environment.

The location of the analogous sections is similar, although in connexin they also include the extra-membrane part—the part exposed on the outside of the cell. In pannexin, this part is not covered by the analogous segments.

Summarizing this part of the analysis it should be stated that the analogous chain segments in pannexin seem to be closer to the T profile, whereas in connexin the status of these segments is clearly different from the idealized profile.

The question arises as to the similarity of the sequences in the segments identified as analogous. It turns out that the sequences of the two proteins show no similarity (as reported earlier) according to the BLAST program.

The determined values of correlation coefficients for the determined segments turn out to be very low (<0.2) and even take values <0.0 for some segments. Similar structural roles are realized in these proteins by means of different hydrophobicity systems.

### 2.4. Status of the Extra-Cellular, Membrane, and Intra-Cellular Parts of Complex

The structure of the connexin and pannexin channel distinguishes between a membrane-anchored part, a part exposed towards the cytoplasm, and a part exposed outside the cell. These zones are designated in Figure 9.

The status of the part of the complex exposed on the outside of the cell membrane was evaluated by the value of the *RD* parameter determined for the beta-sheet and for each fragment of this plate (Table 3). Since this beta-sheet, consisting of three beta-sheet segments, is in both proteins located in a very similar location in the part exposed on the outside of the cell, it was decided to treat this beta-sheet as representing the part exposed on the outside of the cell.

In connexin, the status of all segments as well as the entire beta sheet both within a single chain and within a complex reveals lower maladjustment to the theoretical profiles. For comparative purposes, it is important to analyze the status of the intra-membrane part and fragments exposed to the cytoplasm and outside the cell (Figure 9).

The summary of the results given in Table 3 suggests the similarity of the membrane-anchored fragment of the complex. It may be interpreted as the result of common characteristics: contact with membrane and presence of channel. However, the cytoplasm-directed part seems to be more disordered with respect to idealized hydrophobicity distribution in pannexin for both complex and individual chain status. The part exposed to the external environment appears highly disordered in connexin. The status of this part of both the chain and the complex has a special meaning in the context of generating super-channels composed of two hemi-channels. In the structure of this fragment of the complex, the possibility of interaction with an analogous symmetrical second part is probably encoded, creating a gap junction connecting the membranes of two cells possible.

In the part exposed to the outside, there is a beta-sheet consisting of three sections of Beta-strands. The summary of the values of the RD parameter (Table 4) reveals lower values of RD in individual chain structures; however, from the point of view of the construction of the complex, the status of this Beta-sheet appears to be highly inconsistent with the idealized distribution.

Comparing the status of a single chain as an individual structural unit to a component of the complex is aimed at assessing the possibility of the chain folding in an aqueous environment. It turns out that the values of RD are generally lower for the segments included in the Beta-sheet in the chain structure treated as an individual structural unit.

The part distinguished as “external” requires comparison with the analogical part of connexin able to generate a gap junction in the form of two-hemichannel construction.

## 3. Discussion

Summarizing the obtained results: the degree of similarity of the hydrophobicity profile turns out to be comparable for both channels. The differences are obviously due to the different chain lengths, although the system of local hydrophobicity deficits caused by the presence of the channel and local hydrophobicity redundancies on the surface sections (interaction with the membrane) turns out to be consistent with the predictions for membrane proteins with the channel present in the central part of the complex.

The discussed proteins are further examples of the application of the FOD model and its modified FOD-M version to describe the structures of proteins, including transmembrane proteins in particular. Despite the visible structural similarity, as shown by the profile sets (Figure 4 and Figure 5), the analysis of the corresponding segments shows differences in functionality (Figure 5 and Figure 7). This conclusion is based on the assumption that the hydrophobicity distribution is important as a potential factor for the readiness for stabilization in the hydrophobic environment of the membrane as well as for potential interactions with other molecules, especially in parts exposed outside the membrane (Table 3). This is revealed in the summaries presented in Figure 6 and Figure 9.

The presence of the channel is clearly visible in all analyzes, which is expressed by a significant deficiency of hydrophobicity in the areas of the expected high concentration of hydrophobicity in the central part.

The presence of similarity as well as discrepancies between these two compared transmembrane complexes playing the role of channels may identify the functional differences. The common observation is that the parts of molecules/complexes of similar status in both compared systems accordant with idealized hydrophobicity distribution can play a similar structural role to stabilize the construction. The similarities of the membrane-anchored parts seem to define the role of the environment. However, the difference in external parts may be related to the specificity of molecules transported or the system responsible for the recognition of transported molecules.

From the point of view of the FOD-M model, the M distribution of linear form parallel to the X-axis may symbolize the independence of the water environment. The similarity of M distribution to R distribution represents the status deprived of any specific hydrophobicity distribution except the unified one. The general analysis of this observation is necessary for further analysis. 

Since connexins as well pannexins are responsible for transport they are also responsible for communication on the whole organism level. The dysfunction of these proteins causes serious diseases [50,51,52,53,54,55,56,57,58,59]. The therapy techniques focus the attention on stem cell application [60,61,62].

The conclusions from the presented analysis with respect to the disorders and possible pathological consequences are of a general character. The commonly known is the influence of a mutation on the physiological role of proteins. The mutation introduces a locally or globally different structure (a classic example of sickle cell hemoglobin [63]). Nevertheless, the external conditions in which the folding process takes place turn out to be of equal importance. Application of the discussed model expressing the possible influence of the environmental conditions reveals the importance of the environment on the protein folding process including membrane proteins in particular. The commonly known change in water conditions (such as pH, salt concentration, presence of unusual chemical compounds, etc.) may influence the final protein structure to the extent of changing its role, making the activity impossible or lowering the efficiency of a particular reaction. The differences in COVID-19 pandemic scenarios were shown to be the consequence of traditional dietary customs. Particularly, permanent consumption of low percentage alcohols in the everyday diet may change the characteristics of body fluids to the degree not registered by the organism as pathological. However, the aggressive attack of the virus in changed conditions influencing the protein folding process—the most sensitive one to its environment—makes the defense against the virus of the organism much weaker [64]. It shall be noted that the hard alcohol consumption initiates the degradation processes while the low consumption may not. The organism gets adopted to changed conditions and specificity of the body fluids only for the normalized life scenarios. It was shown that the countries with the traditions of permanent low percentage alcohol consumption revealed significantly higher levels of morbidity as well as mortality in comparison with countries of low alcohol consumption (Far East countries with deficiency of alcohol dehydrogenase activity) [64].

## 4. Materials and Methods

### 4.1. Fuzzy Oil Drop Model

The Fuzzy Oil Drop (FOD) model has already been described many times in the literature, see for example [65]. The full text of the description is available at [65]. The FOD model assumes that a polypeptide chain is composed of amino acids that exhibit the nature of bi-polar molecules that in the aquatic environment tend to generate a micelle-like structure with a centric hydrophobic core. This idealized (theoretical) distribution T can be modeled by a 3D Gaussian function on the protein body. The sequence limitations where the amino acids are joined by the covalent bonds result in the observed distribution O matching the theoretical one to a greater or lesser degree. Let us formally define the two distributions T and O.

The theoretical distribution *T* is defined by the hydrophobicity $H_{i}^{T}$ (i=1,…, N, N being the number of residues) expressed by the value of the 3D Gaussian function at the position of the *i*-th effective atom (i.e., the average position of atoms that make up the *i*-th residue):$$H_{i}^{T}=\frac{1}{H_{sum}^{T}}\exp\left(\frac{-{\left(x_{i}-\bar{x}\right)}^{2}}{2\sigma_{x}^{2}}\right)\exp\left(\frac{-{\left(y_{i}-\bar{y}\right)}^{2}}{2\sigma_{y}^{2}}\right)\exp\left(\frac{-{\left(z_{i}-\bar{z}\right)}^{2}}{2\sigma_{z}^{2}}\right)$$

The values for the $\sigma_{x}$, $\sigma_{y}$, $\sigma_{z}$ parameters are determined based on the molecule under consideration. 

The observed distribution *O* is defined by the hydrophobicity $H_{i}^{O}$ at the position of the *i*-th effective atom according to the Levitt [66]:$$H_{i}^{O}=\frac{1}{H_{sum}^{O}}\underset{j}{\sum }\{\begin{array}{c} \left(H_{i}^{r}+H_{j}^{r}\right)\left(1-\frac{1}{2}\left(7{\left(\frac{r_{ij}}{c}\right)}^{2}-9{\left(\frac{r_{ij}}{c}\right)}^{4}+5{\left(\frac{r_{ij}}{c}\right)}^{6}-{\left(\frac{r_{ij}}{c}\right)}^{9}\right)\right),\;\mathrm{for}\;r_{ij}\leq c\\ 0,\;\mathrm{for}\;r_{ij}>c\; \end{array}$$

The hydrophobicity $H_{i}^{O}$ collects the hydrophobic interactions in distance-dependent form as given in the above formula with the cutoff distance (*c*) according to the original work [66]—9 Å. The $H_{i}^{r}$ and $H_{j}^{r}$ denote the intrinsic hydrophobicity of *i*-th and *j*-th residues. The purpose of denominators $H_{sum}^{T}$ and $H_{sum}^{O}$—being the sum of all $H_{i}^{T}$ and $H_{i}^{O}$ respectively, is to normalize the hydrophobicities to the range 〈0,1〉.

The example of the theoretical *T* (dark blue) and observed *O* (pink) hydrophobicity distribution is presented in Figure 10A.

The *T* and *O* distributions can be quantitatively compared using the divergence entropy $D_{KL}$ between the two distributions *P* and *Q* introduced by Kullback-Leibler [67]
$$D_{KL}\left(P|Q\right)=\overset{N}{\underset{i=1}{\sum }}P_{i}\log_{2}\frac{P_{i}}{Q_{i}}$$
where $P_{i}$—probability observed (in our model—$H_{i}^{O}$, the observed hydrophobicity for the *i*-th residue), $Q_{i}$—reference probability (in our model—$H_{i}^{T}$, the theoretical hydrophobicity for the *i*-th residue).

Next, we introduce the reference distribution R, being the uniform one where *i*-th residue is assigned the same hydrophobicity $R_{i}=1/N$, N being the number of residues in a polypeptide chain (Figure 10B—light blue line). This distribution represents a lack of any variation in the hydrophobicity within a molecule. 

A comparison of the two $D_{KL}$ values, $D_{KL}\left(O|T\right)$ and $D_{KL}\left(O|R\right)$ shows which “distance” is closer. The values $D_{KL}\left(O|T\right)$ less than $D_{KL}\left(O|R\right)$ allow inferring the presence of a centric concentration of hydrophobicity and thus the presence of a hydrophobic core. 

To eliminate the necessity of using the two values, the following parameter RD—Relative Distance is introduced:$$RD=\frac{D_{KL}\left(O|T\right)}{D_{KL}\left(O|T\right)+D_{KL}\left(O|R\right)}$$

The parameter *RD* expresses the degree of adjustment of the hydrophobicity distribution observed in a given structure—resulting from the distribution of residues with a specific intrinsic hydrophobicity to the idealized distribution expressed by a 3D Gaussian function spread over the folding chain at a given moment of the folding process. 

The values of RD<0.5 (being the cut-off value) indicate the presence of the hydrophobic core generated during the folding process. The ideal theoretical hydrophobicity distribution in the protein means the micelle-like state guaranteeing solubility without the possibility of interaction except for random interaction with ions or low molecular weight compounds. The larger deviations of the *O* from the *T* hydrophobicity distribution (i.e., when the cut-off value of 0.5 is exceeded) carry information about the specificity of a given protein, enabling, for example, interaction with a specific ligand by the appropriate adjustment of the interaction field. Of course, it is also possible to bind the polar ligand on the protein surface without disturbing the structure of the hydrophobic core.

The modification of the FOD model, the so-called FOD-M model [42], extending the participation of a non-polar environment in protein folding relies on introducing the structural specificity of membrane proteins—including membrane proteins serving as an ion channel [42,43,44].

Following the hydrophobicity distribution in membrane proteins (where an exposure of hydrophobic residues is expected on the surface and the presence of polar ones—in the center), we define the modified hydrophobicity distribution M which is “inverted” to the centric theoretical distribution *T* and can be expressed by the function:$$M_{i}=T_{MAX}-T_{i}$$
where $T_{MAX}$ is the maximum value in the theoretical distribution *T*.

The distribution *T* is modified, assigning to individual residues a status in the form of complement to the value expected for the centric distribution. However, it turns out that the omnipresence of the aquatic environment also imprints the structure of the membrane protein. Therefore, the external field directing the protein folding process turns out to be a consensus between the centric field and the inverted one, and can be expressed as:$$M_{i}={\left[T_{i}+{\left(T_{MAX}-T_{i}\right)}_{n}\right]}_{n}$$
where the index *n* denotes normalization which relies on dividing each element (i.e., the partial hydrophobicity from *i*-th residue) of the set by the sum of all elements in it. After normalization, the sum of all elements is equal to 1.

The *M* distribution expresses the influence of the membrane environment in the extreme case, which is the membrane, being the fully hydrophobic environment. The coefficient *K* was additionally introduced to make the definition of a distribution *M* more universal:$$M_{i}={\left[T_{i}+K{\left(T_{MAX}-T_{i}\right)}_{n}\right]}_{n}$$

The coefficient *K* expresses the consensus between the water environment (centric hydrophobic core) and the hydrophobic environment of the membrane (or the presence of any hydrophobic compound modifying the idealized distribution expressed by the 3D Gauss function). Values of the coefficient *K* close to 0 represent proteins with a high degree of centric hydrophobicity while those close to 1—represent structures with a significant part of a membrane environment. It also turns out that the value of a parameter *RD* is highly correlated with the value of coefficient *K*. Both these values express the degree of deviation from the micelle-like hydrophobicity distribution within the protein. The value of parameter *RD* represents the difference from the centric distribution while the value of coefficient *K*—measures the participation of other than polar factors influencing the folding process.

The sample plots of distribution *M* for the three values of coefficient *K* (*K* = 0.5; 1.0; 1.5) are presented in Figure 11A. Figure 11B shows the plot of distribution *M* with a very high value of *K* (*K* = 3) which completely eliminates the presence of a maximum, introducing a minimum in its place. Such situations are observed in ion channels. 

Next, the optimal value of coefficient *K* is determined by seeking the value of *K* corresponding to the smallest value $D_{KL}\left(O|M\right)$ of the distance between the two distributions: observed *O* and membrane *M*. For such optimal value of coefficient *K* (see Figure 12).

The parameter *RD* expressing the relative distance between the distributions *O* and *T* is supplemented with the parameter *RD* calculated for the relative distance between the distributions *O* and $M_{opt}$ (the distribution *M* corresponding to the optimal value of a coefficient *K* as described above):$$RD_{Kopt}=\frac{D_{KL}\left(O|T\right)}{D_{KL}\left(O|T\right)+D_{KL}\left(O|M_{opt}\right)}$$

### 4.2. Programs Used

The potential used has two possible access to the program:

The program allowing calculation of RD is accessible upon request on the CodeOcean platform: https://codeocean.com/capsule/3084411/tree, accessed on 15 June 2022. Please contact the corresponding author to get access to your private program instance.

The application—implemented in collaboration with the Sano Centre for Computational Medicine (https://sano.science, accessed on 26 April 2022) and running on resources contributed by ACC Cyfronet AGH (https://www.cyfronet.pl, accessed on 26 April 2022) in the framework of the PL-Grid Infrastructure (https://plgrid.pl, accessed on 26 April 2022)—provides a web wrapper for the abovementioned computational component and is freely available at https://hphob.sano.science, accessed on 26 April 2022.

The VMD program was used to present the 3D structures [68,69].

### 4.3. Calculation Procedure

The applied calculation procedure is to determine the status of the chain, which includes the section identified as IDR. The characteristic is given by the RD parameter for the T-O-R relation and the value of the K parameter, which determines the degree of the proportion of a factor other than the aquatic environment. Additionally, the status of this segment IDR as a component of the structural unit is determined. If the chain has a domain structure, the structural unit against which the status of the IDR is determined is precisely the domain.

The assessment of a structural unit (chain/domain) from the point of view of the fuzzy oil drop model consists in generating a 3D Gaussian function encapsulating the entire unit. The value of the RD parameter determined with it defines the status as a whole. On the other hand, the status of a section classified as IDR consists in determining the contribution of a given section to the construction of the entire structural unit. In this situation, the selected fragment of T and O profiles, which was obtained for the entire unit, is subjected to normalization and the value of RD is determined. This value determines the share of a given segment in the structure of the centric hydrophobic core. For the normalized fragments of the T and O profiles, the optimal K value is determined, which determines the degree of modification necessary to determine the status of a given fragment of the chain. The classification is common: RD < 0.5 both for the entire structural unit and for the selected section means participation in the structure of the hydrophobic core. Otherwise, the given unit does not show the presence of a hydrophobic core, and the section with such characteristics is treated as disturbing the system expected for the hydrophobic core.

## 5. Conclusions

The use of the FOD model and its modified FOD-M version enables the identification of similarities and differences, both structural and functional. In the FOD model, we assume that the distribution of hydrophobicity is a record of the stabilization of the structure (hydrophobic core) and potential interactions (the local hydrophobicity deficit expresses the readiness to interact with other molecules, and the local excess is the readiness to interact with hydrophobic systems, including the membrane). A significant deficit of hydrophobicity at the site of the expected hydrophobic core is the identification of the channel’s presence. A high level of hydrophobicity over almost the entire surface means preparation for interaction with the membrane. The differentiation of the sections with the O profile comparable to the T profile means that the stabilization centers are adapted to the structure of a given protein. The compatibility of these two profiles is a factor of stabilization resulting from a balanced system resulting from intramolecular interactions and external factors.

The FOD-M model turns out to be universal. The soluble proteins are described by 0 < K < 0.4. The values of 0.4 > K > 0.0 in globular proteins prove a different degree of influence on the structuring of proteins of external factors. The identification of specificity identified in a periplasmic protein expressed by K = 0.6 also suggests the possibility of the universal character of the FOD-M model [70].

The comparison of the currently analyzed proteins with other proteins active in the membrane environment reveals their significant differentiation expressed, for example, by the value of the K parameter from relatively low such as K = 0.6 to the value K > 3 [42,43,44]. Therefore, the analysis of other groups of membrane proteins is carried out.

## Figures and Tables

**Figure 1 biomedicines-10-01504-f001:**
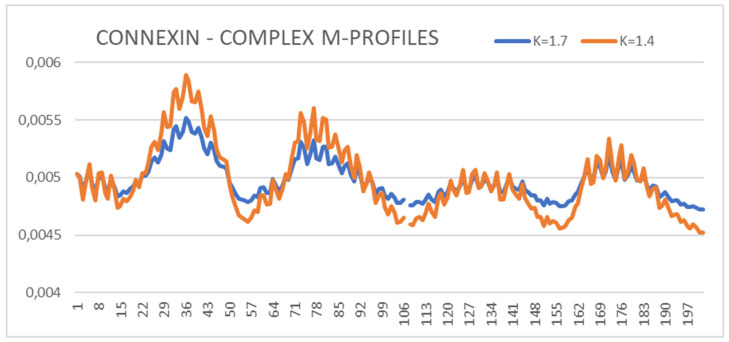
The summary of M distributions for connexin in the presence of Ca^2+^ (K = 1.7) and in its absence (K = 1.4)—status of chain A in complexes is shown. The numbers on the x-axis represent the order of residues in a chain—the numbers are not the positions as it is given in PDB.

**Figure 2 biomedicines-10-01504-f002:**
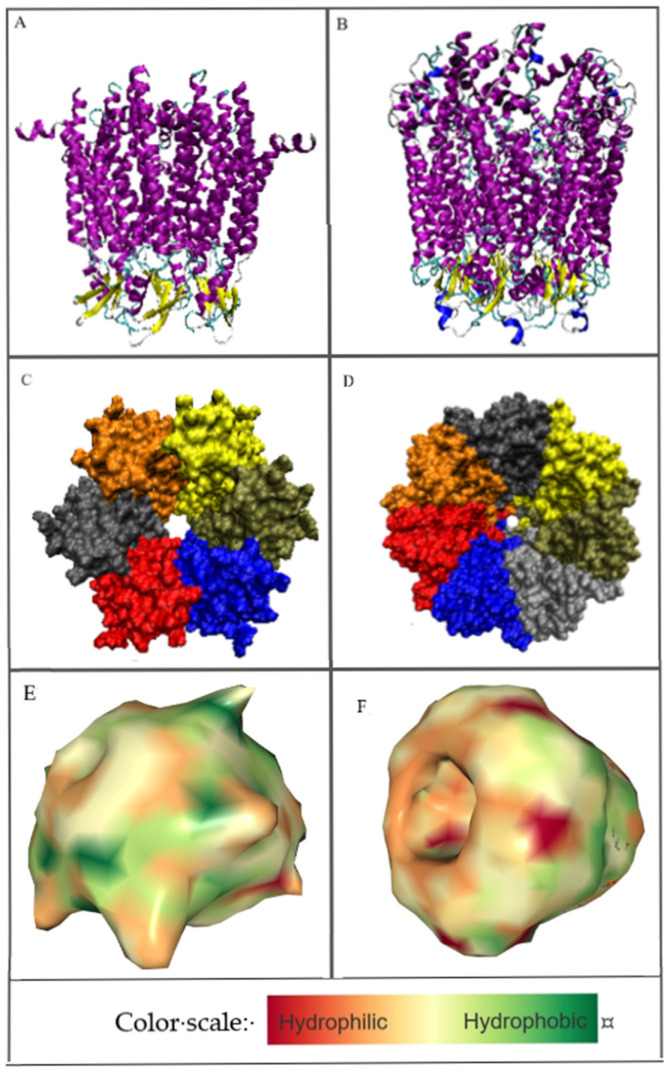
3D structure of: (**A**)—connexin’s hemi-channel, (**B**)—pannexin’s channel with secondary structure distinguished: yellow—Beta-sheet, magenta—transmembrane helices, dark blue—external helices, green—random coil. The beta-sheet is in the outer membrane part that is exposed to the outside of the cell. (**C**,**D**) represent the differences in channel construction as well as the six and 7-fold symmetry. (**E**,**F**)—hydrophobicity visualization: connexin and pannexin respectively. Images (**E**,**F**) created by NGL Viewer: Rose et al. (2018) NGL viewer: web-based molecular graphics for large complexes. Bioinformatics doi:10.1093/bioinformatics/bty419, and RCSB PDB. (**A**,**B**)—orientation of the molecules: Top—intra-cellular area, bottom—extra-cellular area; (**C**,**D**)—orientation from the point of view of the intra-cellular area—symmetry axis perpendicular to the plane of the figure; (**E**,**F**)—side view arbitrarily selected to visualize the shape differences of the compared complexes.

**Figure 3 biomedicines-10-01504-f003:**
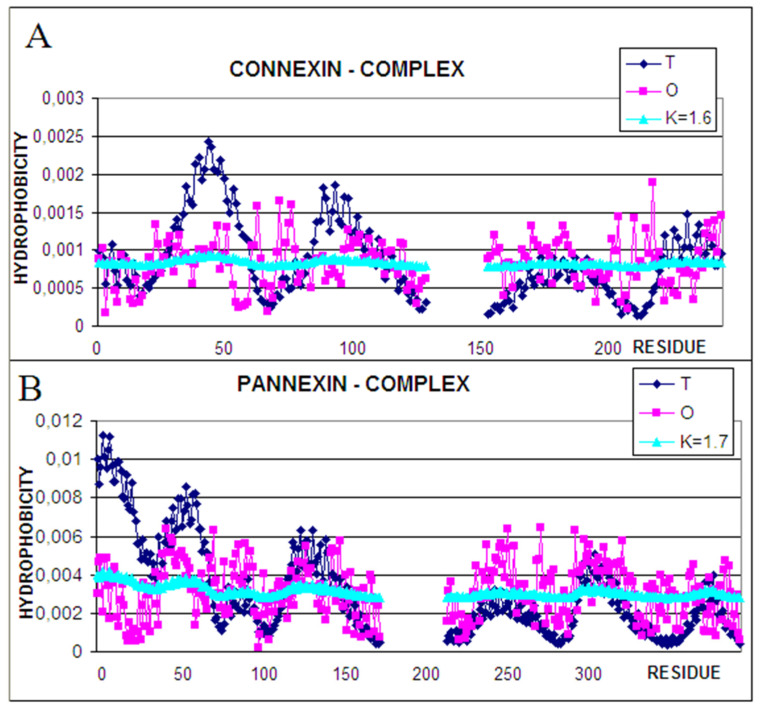
Profiles for the distribution of T (navy), O (pink), and M (turquoise)for K values as shown in the legend: (**A**)—6L3T—connexin, (**B**)—7F8J—pannexin.

**Figure 4 biomedicines-10-01504-f004:**
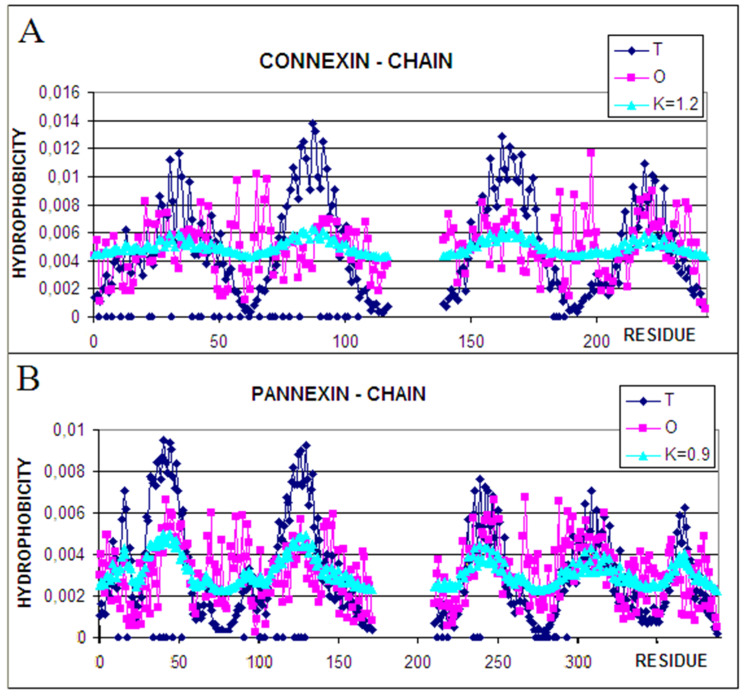
Profiles for the T (dark blue), O (pink), and M (turquoise) distributions were calculated for single chain (**A**)—connexin, (**B**)—pannexin. Blue dots—bottom line—residues interacting with adjacent chains.

**Figure 5 biomedicines-10-01504-f005:**
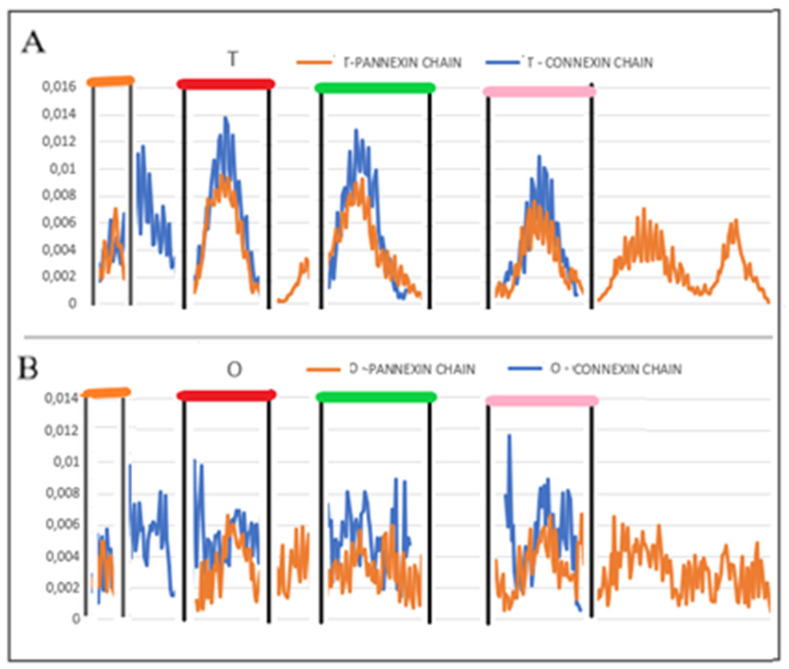
Summary of profiles for identification of analogous segments in pannexin (red) and connexin chains (blue) (**A**)—T; (**B**)—O; for chains treated as individual structural units pannexin (blue) and connexin (orange) and sections identified as analogous in the structure of single chains—identification based on the T profile. The colors differentiate the fragments as shown also in Figure 6 and Table 2.

**Figure 6 biomedicines-10-01504-f006:**
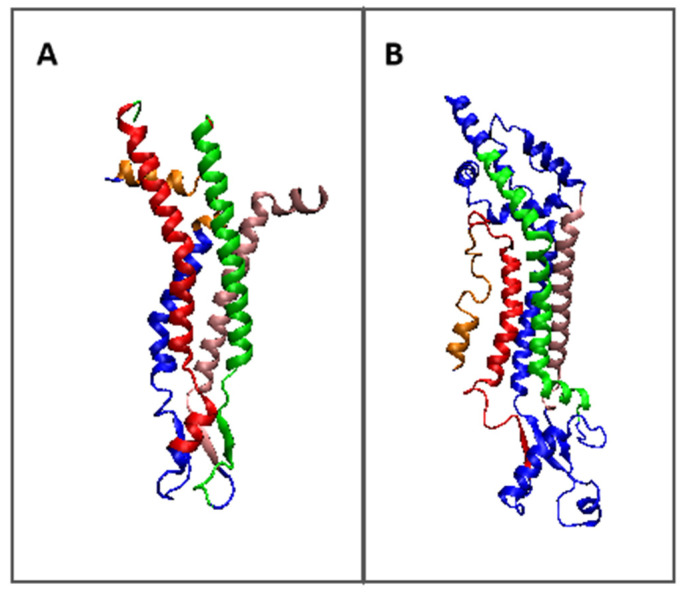
3D presentation showing the analogous fragments in the chain treated as individual structural units: (**A**)—connexin, (**B**)—pannexin. Colors as given in Table 2 and Figure 5.

**Figure 7 biomedicines-10-01504-f007:**
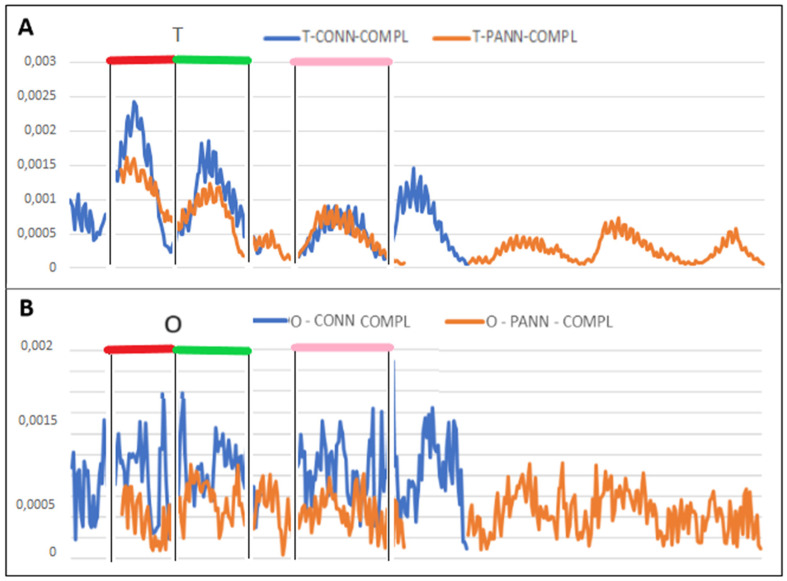
Comparison of hydrophobicity profiles: (**A**)—profile T, (**B**)—profile O analogical in pannexin (red) and connexin (blue) as observed in the complex and sections identified as analogous in *T* profiles of complexes. Similar fragments are distinguished by colors according to the presentation in Figure 8 and Table 2.

**Figure 8 biomedicines-10-01504-f008:**
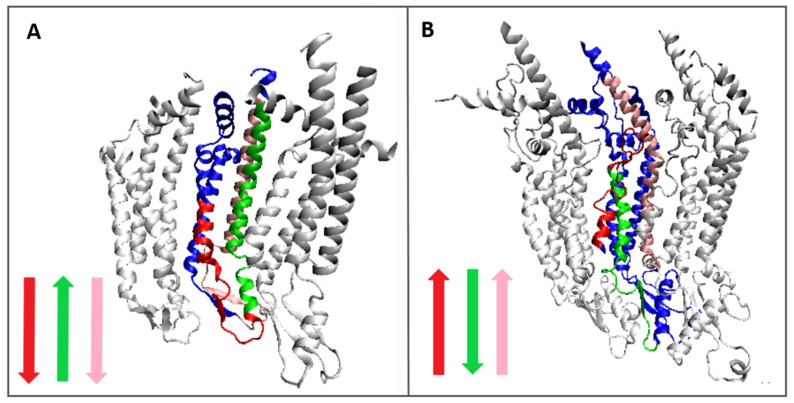
3D presentation showing the analogous fragments in the complex—(**A**)—connexin, (**B**)—pannexin. Colors identification is given in Table 2. Notice the opposite orientation of the selected fragments shown by color arrows. Chain (**A**)—dark blue. The two white chains in each structure visualize the surrounding of the discussed chain. Molecule orientation:—Top—intra-cellular; Bottom—extracellular direction. Colors as given in Table 2 and Figure 7.

**Figure 9 biomedicines-10-01504-f009:**
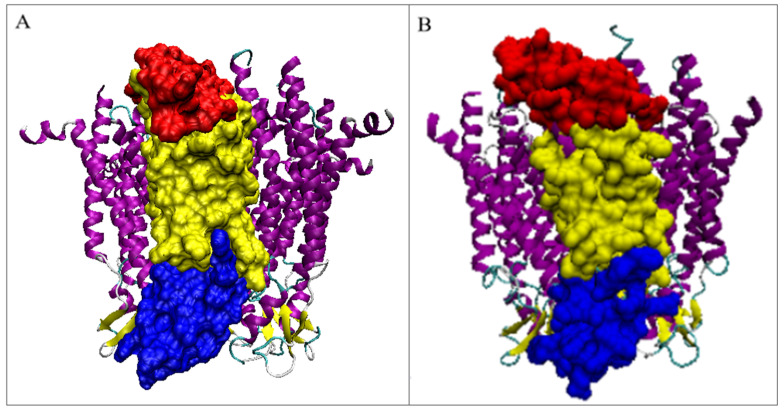
3D presentation: (**A**)—pannexin; (**B**)—connexin with highlighted parts: the part facing the cytoplasm—red, the part anchored in the membrane—yellow, the part facing the outside of the cell—blue marine. Violet chains—adjacent chains to visualize the mutual orientation of adjacent chains. This status is given in Table 3.

**Figure 10 biomedicines-10-01504-f010:**
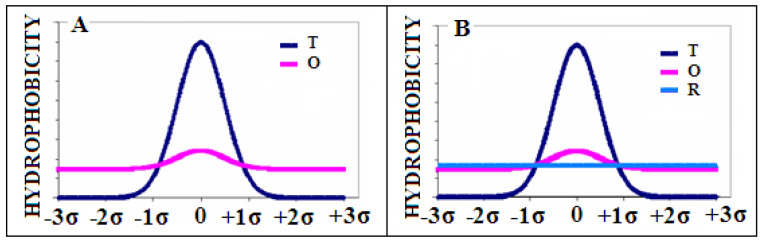
Examples of distributions: (**A**)—distribution O (pink) and reference Gaussian distribution T (with parameter σ) expressing the presence of the central hydrophobic core (dark blue). (**B**)—the second reference distribution R (light blue) superimposed—expressing a lack of any variation in the hydrophobicity.

**Figure 11 biomedicines-10-01504-f011:**
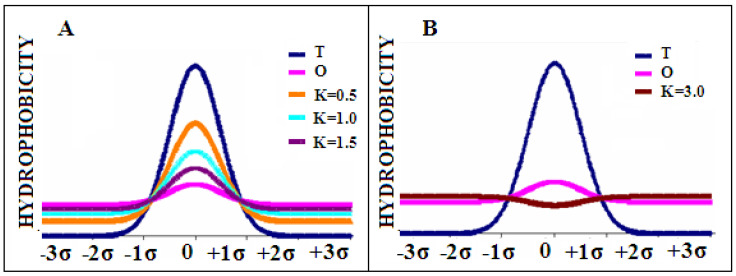
(**A**)—the comparison of plots of distributions *M* for the three different values of *K*; (**B**)—plot of distribution *M* for a very high value of *K* (3.0). The distributions *T* and *O* are marked in navy blue and pink respectively.

**Figure 12 biomedicines-10-01504-f012:**
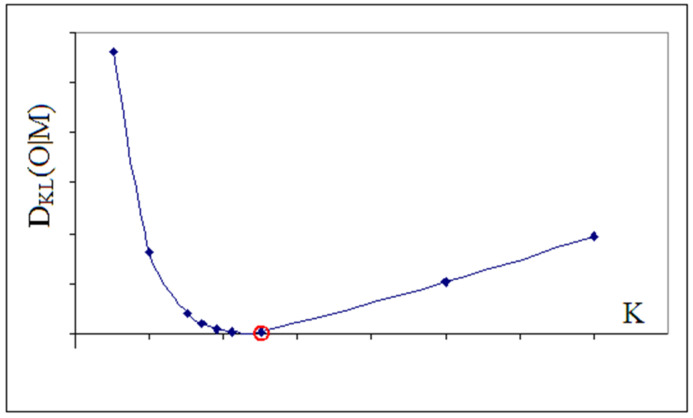
Determining the optimal value of coefficient *K*. The lowest $D_{\mathrm{KL}}$ value is marked (red circle).

**Table 1 biomedicines-10-01504-t001:** The summary of the values of parameters RD and K for the selected structural units. P-P—the residues involved in cross-chain interactions, No P-P—the remainder of the protein excluding the residues involved in P-P, SS—disulfide bonds. P-Ligand—status or residues engaged in ligand interaction, No P-Ligand—the status of the molecule with residues engaged in interaction with ligand eliminated from the calculation. The results are shown for complete complexes treated as structural units (3D Gauss function spanned on the complete complexes as well as for single-chain treated as an individual structural unit—the 3D Gauss function spanned on a single chain).

Complex or Chain	Structural Unit	Pannexin 7 Chains		Connexin 6 Chains	
		RD	K	RD	K
Complex	Channel/hemichannel	0.738	1.7	0.755	1.6
	SS	0.627	0.8	0.761	1.3
		0.611	0.7	0.756	1.2
				0.772	1.1
	P-P	0.597		0.625	
	No P-P	0.745		0.768	
	P-Ligand	0.596		0.625	
	No Ligand	0.745		0.768	
Chain A	individual unit	0.696	0.9	0.777	1.2
	SS	0.735	1.0	0.852	1.4
		0.604	0.6	0.847	1.1
				0.848	0.9
	P-P	0.821		0.646	
	No P-P	0.652		0.778	
	P-Ligand	0.542		0.646	
	No Ligand	0.702		0.778	

**Table 2 biomedicines-10-01504-t002:** Comparison of *RD* and *K* values for analogous sections determined on the basis of similarity of Figure 5. D Gauss function spanned on a single chain) and on the basis of *T* profiles for chains representing complexes (all chains in the complex represent the same status). 3D Gauss function spanning the entire complex.

	Pannexin			Connexin		
	Fragment	RD	K	Fragment	RD	K
Chain Iindividual						
orange	2–21	0.713	1.5	2–21	0.571	0.5
red	22–68	0.649	0.5	61–107	0.866	1.1
green	96–146	0.607	0.5	127–177	0.862	1.1
pink	202–245	0.453	0.3	178–221	0.617	0.7
Complex						
red	2–31	0.481	0.1	30–59	0.690	0.5
Green	32–70	0.706	0.5	60-98	0.785	0.9
pink	99–148	0.594	0.4	127–176	0.740	0.7

**Table 3 biomedicines-10-01504-t003:** The status of the complex part and the single chain is defined as cytoplasmic—the extra-membrane part facing the cytoplasm of the cell, membrane—part anchored to the membrane; external—the part exposed outside the cell.

	Connexin				Pannexin			
	Complex		Chain		Complex		Chain	
Fragment	RD	K	RD	K	RD	K	RD	K
Cytoplasmic	0.717	1.2	0.703	1.4	0.718	1.7	0.754	1.7
Membrane	0.816	1.4	0.747	0.7	0.717	1.3	0.553	0.4
External	0.748	3.2	0.752	2.1	0.704	1.7	0.657	1.0

**Table 4 biomedicines-10-01504-t004:** Status of the beta-structure segments and the entire beta-sheet in the two compared proteins.

Connexin			Pannexin		
	Complex	Chain		Complex	Chain
Fragment	RD	RD	Ffagment	RD	RD
50–54	0.741	0.705	63–67	0.763	0.651
165–169	0.718	0.560	244–246	0.739	0.777
178–183	0.783	0.595	263–268	0.687	0.597
β-SHEET	0.770	0.648	Β-SHEET	0.787	0.615

## Data Availability

All data can be available on request addressed to corresponding author. The program allowing calculation of RD is accessible on GitHub platform: https://github.com/KatarzynaStapor/FODmodel and on platform https://hphob.sano.science, accessed on 26 April 2022.

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
