# Peer review of "Connexins and Pannexins—Similarities and Differences According to the FOD-M Model"

_biomedicines, 2022, doi:10.3390/biomedicines10071504_

Round 1

Reviewer 1 Report

The manuscript by Roterman  et al. reported an analysis of connexins and pannexins using FOD-M model. Despite the sequences of connexin and pannexin are not similar, they have comparable profiles by FOD-M model. The authors provided another application to the FOD-M model. The current study is on a topic of relevance to the readers of the journal, especially for the “Connexins and Pannexins in Embryonic and Fetal Development” special issue. I found the writing of this paper can be improved. Please see the comments below.

Major comments:

  1. Please paraphrase the sentences for the Introduction. Don’t use the exact sentences from the references or online resources. For example, the first two sentences “A transmembrane protein is a type of integral membrane protein that spans the entirety of the cell membrane. Many transmembrane proteins function as gateways to permit the transport of specific substances across the membrane” is exactly what Wikipedia said about transmembrane proteins.

The sentence “Pannexins primarily form oligomeric structures embedded in a single plasma membrane that when open, provide a conduction pathway between the cytosol and extracellular space.” in the Introduction is worded exactly the same as a sentence in the paper “Pannexin channels are not gap junction hemichannels” by Gina Sosinsky et. al. on Channel in 2011. I did not see that paper cited as a reference here.

  1. After briefly talking about the definition and function of transmembrane protein, the author started to talk about the hydrophobicity distribution of membrane proteins and soluble proteins from line 7 to line19. The sentences are not well written and hard to read. The author might want to add bullet points to guide the readers. For example, the author could write something like “There are several key factors contributing to the difference of hydrophobicity distribution observed from membrane proteins and soluble proteins:1) the polar aquatic environment and hydrophobic membrane environment... 2) the folding process which is entropy-driven...”
  2. It’s better to move the challenges for membrane protein research to a new paragraph. Also, since the author mentioned the natural environment, it is better to elaborate on some recent research results here. Besides using detergent to stabilize membrane proteins, researchers use lipid to stabilize membrane proteins for both structural and functional papers (keywords for those papers: lipid nanodisc, liposomes). Also, besides using CryoEM to study the structures of membrane protein, there are X-Ray, Solid-state NMR routinely used these days. 
  3. The introduction lacks a literature review on connexins/pannexins. I know this special issue probably has several good reviews on this topic already. Just a brief review here can make the introduction more complete. How do researchers stabilize connexins/pannexins to form channels? Are there any high-resolution structures available? If not, are there homolog structures (such as innexins) available? Are they important for diseases?
  4. Are there any methods other than FOD? Please give a brief literature review in the introduction before jumping into FOD-M.
  5. Why connexins from sheep and pannexins from human? The author mentioned the connexins from sheep and pannexins from human showed no sequence similarity. Did the author perform a sequence alignment between connexins from sheep and connexin from human?
  6. For Results section 2.1, an overall description of the 3D structures of connexins/pannexins channel would be nice. What methods they used to get the structures? Were the sequences missing because the researchers mutated to enhance expression, or were they both wild types but had great flexibility so no electron density resolved? Where are the missing sequences in term of zone (transmembrane zone or extracellular zone)? Were there intrasubunit/intersubunit interactions observed (especially since the authors mentioned P-P interaction later)? Also, the author should mention the overall picture first instead of jumping from point to point for similarities and differences. For example, the author should mention 6 subunits in the connexin channel and 7 in pannexin channel before mentioning beta sheets (the author mentioned it is 6 or 8 subunits for pannexin channel, why it is 7 here?)
  7. It would be interesting to include a not-so-similiar channel such as innexin or even an ion channel in the analysis to see the power of FOD-M analysis and to further validate the analysis.

Minor comments:

  1. Line3 of Abstract, “they role’ should be ‘their role’.
  2. The contents of the abstract are great, but the expressions can be improved. For example, the first three sentences can be combined into one sentence.
  3. Please write the full name for FOD model (‘Fuzzy oil drop model’) when it appears for the first time in the abstract.
  4. Line 6 of Introduction, please clarify if it’s the conformational change of membrane proteins that is frequently required.
  5. In the “Connexin” part of the Introduction:

5.1 ”assemble to form”->”form”;

5.2 ”Each gap junction is composed of two hemichannels”-> ”Each gap junction is composed of two hemichannels head-to-head” or “Each gap junction is composed of two hemichannels, which are located in the plasma membrane of adjacent cells”;

  1. In the “Pannexin” part of the Introduction: Although letting the readers know how many subunits of pannexins can form a channel is relevant, it surprises me that it is the only sentence for pannexin introduction in the entire paper. Since the author mentioned connexins are in vertebrates, it’s worth mentioning pannexins are also vertebrate proteins here. Please also see point 4 in major comments and extend the introduction for connexin & pannexin.
  2. In the last paragraph of the introduction, ”the object of analysis...are” should be “the objects”.
  3. Figure 1 shows the 3D models of connexins/pannexins channel. Currently, there are yellow, blue, cyan, and magenta, and the authors did not use these color codes in the text. They should mention, for example, the beta sheets in the extracellular zone are colored yellow. Also, it might be nice to have hydrophobicity/electrostatic potential surface mapped to those models in Figure1 as well. Those can be easily done with a PDB model.
  4. In the Material and Method section, “The Fuzzy Oil Drop (FOD) model has already been described many times in the literature” did not give useful information. The author could either mention when FOD first appeared or many different applications of FOD here instead.
  5. Figure numbers did not match. Figure 11 is referred to as Figure12 in the text. There is no Figure 10.

Author Response

Reviewer I

Dear Reviewer

Many thanks for insipring comments. We did our best to follow your advices.

We hope – the current version will be found as satisfactory.

The detailed information given below.

All changed and newly inytroduced frafments are given in red.

Sincerely yours;

Irena Roterman

Yes

Can be improved

Must be improved

Not applicable

Does the introduction provide sufficient background and include all relevant references?

( )

( )

(x)

( )

Are all the cited references relevant to the research?

(x)

( )

( )

( )

Is the research design appropriate?

( )

(x)

( )

( )

Are the methods adequately described?

(x)

( )

( )

( )

Are the results clearly presented?

(x)

( )

( )

( )

Are the conclusions supported by the results?

( )

(x)

( )

( )

Comments and Suggestions for Authors

The manuscript by Roterman  et al. reported an analysis of connexins and pannexins using FOD-M model. Despite the sequences of connexin and pannexin are not similar, they have comparable profiles by FOD-M model. The authors provided another application to the FOD-M model. The current study is on a topic of relevance to the readers of the journal, especially for the “Connexins and Pannexins in Embryonic and Fetal Development” special issue. I found the writing of this paper can be improved. Please see the comments below.

Major comments:

  1. Please paraphrase the sentences for the Introduction. Don’t use the exact sentences from the references or online resources. For example, the first two sentences “A transmembrane protein is a type of integral membrane protein that spans the entirety of the cell membrane. Many transmembrane proteins function as gateways to permit the transport of specific substances across the membrane” is exactly what Wikipedia said about transmembrane proteins.

The sentence “Pannexins primarily form oligomeric structures embedded in a single plasma membrane that when open, provide a conduction pathway between the cytosol and extracellular space.” in the Introduction is worded exactly the same as a sentence in the paper “Pannexin channels are not gap junction hemichannels” by Gina Sosinsky et. al. on Channel in 2011. I did not see that paper cited as a reference here.

Sorry for this. I usually pass the text of the paper prepared in polish to the translator. Probably the perfect expression in English takes the form present in other documents. Both sentences got changed. The mentioned reference got added to the list of cited literature. Many thanks for this.

  1. After briefly talking about the definition and function of transmembrane protein, the author started to talk about the hydrophobicity distribution of membrane proteins and soluble proteins from line 7 to line19. The sentences are not well written and hard to read. The author might want to add bullet points to guide the readers. For example, the author could write something like “There are several key factors contributing to the difference of hydrophobicity distribution observed from membrane proteins and soluble proteins:1) the polar aquatic environment and hydrophobic membrane environment... 2) the folding process which is entropy-driven...”

Many thanks for this suggestion.

  1. It’s better to move the challenges for membrane protein research to a new paragraph. Also, since the author mentioned the natural environment, it is better to elaborate on some recent research results here. Besides using detergent to stabilize membrane proteins, researchers use lipid to stabilize membrane proteins for both structural and functional papers (keywords for those papers: lipid nanodisc, liposomes). Also, besides using CryoEM to study the structures of membrane protein, there are X-Ray, Solid-state NMR routinely used these days. 

This issue has been completed –new references added.  

  1. The introduction lacks a literature review on connexins/pannexins. I know this special issue probably has several good reviews on this topic already. Just a brief review here can make the introduction more complete. How do researchers stabilize connexins/pannexins to form channels? Are there any high-resolution structures available? If not, are there homolog structures (such as innexins) available? Are they important for diseases?

New positions in references added.

Innexins got mentioned in the discussion. This type of protein is the object of comparable analysis with connexins in our current analysis.

  1. Are there any methods other than FOD? Please give a brief literature review in the introduction before jumping into FOD-M.

The aim of presented materia lis the applicability of FOD model to different proteins acting in different environmental conditions. The influence of environemnt specificity is expressed by K parameter in FOD-M model. This is the main hypothesis of the discussed model.

  1. Why connexins from sheep and pannexins from human? The author mentioned the connexins from sheep and pannexins from human showed no sequence similarity. Did the author perform a sequence alignment between connexins from sheep and connexin from human?

Sheep’s protein got substituted by human protein.

  1. For Results section 2.1, an overall description of the 3D structures of connexins/pannexins channel would be nice.

The description added to the text.

What methods they used to get the structures?

This information added to the text.

Were the sequences missing because the researchers mutated to enhance expression, or were they both wild types but had great flexibility so no electron density resolved?

This examples of structures followed the regulation addressed to crystallographers as to the criterion for structure recognition.

  1. Where are the missing sequences in term of zone (transmembrane zone or extracellular zone)?

This information added – the missing fragment are localised in plasma-directed external part of discussed complexes.

  1. Were there intrasubunit/intersubunit interactions observed (especially since the authors mentioned P-P interaction later)?

The residues identified as interacting between chains are recognised according to available information in PDB data base.

  1. Also, the author should mention the overall picture first instead of jumping from point to point for similarities and differences. For example, the author should mention 6 subunits in the connexin channel and 7 in pannexin channel before mentioning beta sheets (the author mentioned it is 6 or 8 subunits for pannexin channel, why it is 7 here?)

The general description has been added.

  • It would be interesting to include a not-so-similiar channel such as innexin or even an ion channel in the analysis to see the power of FOD-M analysis and to further validate the analysis.

FOD model and its modification has been applied to many membrane proteins. It was applied also to amyloid proteins starting abviuosly from soluble proteins. The power of FOD model is also expressed by the possibility to identify the Protein-Protein interface in complexation [Dygut J, Kalinowska B, Banach M, Piwowar M, Konieczny L, Roterman I. Structural Interface Forms and Their Involvement in Stabilization of Multidomain Proteins or Protein Complexes. Int J Mol Sci 2016 ;17(10):1741. doi: 10.3390/ijms17101741. ]. as well as identification of ligand/substrate binding cavity [Banach M, Konieczny L, Roterman I – Ligand binding site recognition – In Protein folding in silico Woodhead Publishing – currently Elsevier – 2012 pp. 80-94

Banach M, Konieczny L, Roterman I. Use of the fuzzy oil drop model to identify the complexation area in protein homodimers. In Protein folding in silico Woodhead Publishing – currently Elsevier – 2012 pp. 95-122.

The large spectrum of applicability of FOD model is shown in : From globular proteins to amyloids : https://www.sciencedirect.com/book/9780081029817/from-globular-proteins-to-amyloids.

All these positions have been added to the list of references.

Minor comments:

  1. Line3 of Abstract, “they role’ should be ‘their role’. CORRECTED
  2. The contents of the abstract are great, but the expressions can be improved. For example, the first three sentences can be combined into one sentence. CORRECTED
  3. Please write the full name for FOD model (‘Fuzzy oil drop model’) when it appears for the first time in the abstract. CORRECTED
  4. Line 6 of Introduction, please clarify if it’s the conformational change of membrane proteins that is frequently required CORRECTED
  5. In the “Connexin” part of the Introduction:

5.1 ”assemble to form”->”form”;   CORRECTED

5.2 ”Each gap junction is composed of two hemichannels”-> ”Each gap junction is composed of two hemichannels head-to-head” or “Each gap junction is composed of two hemichannels, which are located in the plasma membrane of adjacent cells”;

  1. In the “Pannexin” part of the Introduction: Although letting the readers know how many subunits of pannexins can form a channel is relevant, it surprises me that it is the only sentence for pannexin introduction in the entire paper. Since the author mentioned connexins are in vertebrates, it’s worth mentioning pannexins are also vertebrate proteins here. Please also see point 4 in major comments and extend the introduction for connexin & pannexin. This subject extended.
  2. In the last paragraph of the introduction, ”the object of analysis...are” should be “the objects”. CORRECTED
  3. Figure 1 shows the 3D models of connexins/pannexins channel. Currently, there are yellow, blue, cyan, and magenta, and the authors did not use these color codes in the text. They should mention, for example, the beta sheets in the extracellular zone are colored yellow. CORRECTED
  4. Also, it might be nice to have hydrophobicity/electrostatic potential surface mapped to those models in Figure1 as well. Those can be easily done with a PDB model.

This presentation added.

  1. In the Material and Method section, “The Fuzzy Oil Drop (FOD) model has already been described many times in the literature” did not give useful information. The author could either mention when FOD first appeared or many different applications of FOD here instead.

The full text of the book From globular proteins to amyloids is available on web site https://www.sciencedirect.com/book/9780081029817/from-globular-proteins-to-amyloids. The complete description of the model can be found there.

  1. Figure numbers did not match. Figure 11 is referred to as Figure12 in the text. There is no Figure 10. CORRECTED

Reviewer 2 Report

General Notes: CONNEXINs and PANNEXINs – similarities and differences according to the FOD-M model    Irena Roterman1,*, Katarzyna Stapor2, Piotr Fabian3 and Leszek Konieczny4

INTRODUCTION

Introduction needs editing for clarity and syntax.  The authors have made an admirable effort to provide background information that is accessible to a broad audience. 

Paragraph 1 is particularly difficult to parse and needs to be re-evaluated and re-written for clarity, syntax, and grammar.

Paragraph 2 requires references for pannexin channel formation.  Again, the choices for explaining gap junction formation to a general audience are not clear and would require only a small bit of “tweaking”

For example:                                                                                        Connexins are transmembrane proteins that assemble to form cell-cell channels called gap junctions. Each gap junction is composed of two hemichannels, each of which is made of six connexin subunits. Pannexins are also channel membrane proteins and are formed by six or eight monomers. The way connexins and pannexins form structures within the membrane differ. Connexins, unlike pannexins, provide a mechanism for transport of ions and small signaling molecules between apposed cells. Pannexins have not been shown to form functional cell-cell channels.  Pannexins primarily  form oligomeric structures embedded in a single plasma membrane that when open, provide a conduction pathway between the cytosol and extracellular space.

 Paragraph 3 and 4 also require editing for clarity. 

Why choose sheep Cx46 and human PANX1 to test the model?  Why decide to only analyze the hemichannel structure of Cx46?  Pannexins are believed to only form hemichannels, while the hemichannel structure-function of various connexins is a separate area of study from cell-cell channels.

 RESULTS

A brief explanation of FOD-M and the various analytic features would be helpful.  While a detailed explanation, including derivation of the various elements, is provided in the methods section, acronym definitions in the body of the text are useful for the reader.  This is a manuscript directed towards an educated general audience. To best evaluate the strength of the authors’ conclusions, analysis parameters should be detailed along with presentation of the data.

Fig 1  Figure legend does not provide enough detail identifying the different parts of the structure, including regions of similarity and difference.  

The authors conclude that the T, O, and M distributions are typical for membrane proteins as a class but fail to provide references or general data to support this claim.

Labeling using Pannexin and Connexin (as in Fig 3) would also be helpful.

 Section 2 provides reasonable explanations the “what” and “why” of the analyses presented in the next sections.

 DISCUSSION

The discussion is adequate, but could use some editing for clarity, to adjust awkward phrasing, and fix minor grammatical errors.

Author Response

Reviewer II

Dear Reviewer

Many thanks for insipring comments. We did our best to follow your advices.

We hope – the current version will be found as satisfactory.

The detailed information given below.

All changed and newly inytroduced frafments are given in red.

Sincerely yours;

Irena Roterman

Yes

Can be improved

Must be improved

Not applicable

Does the introduction provide sufficient background and include all relevant references?

( )

( )

(x)

( )

Are all the cited references relevant to the research?

(x)

( )

( )

( )

Is the research design appropriate?

( )

(x)

( )

( )

Are the methods adequately described?

(x)

( )

( )

( )

Are the results clearly presented?

( )

(x)

( )

( )

Are the conclusions supported by the results?

(x)

( )

( )

( )

Comments and Suggestions for Authors

General Notes: CONNEXINs and PANNEXINs – similarities and differences according to the FOD-M model    Irena Roterman1,*, Katarzyna Stapor2, Piotr Fabian3 and Leszek Konieczny4

INTRODUCTION

Introduction needs editing for clarity and syntax.  The authors have made an admirable effort to provide background information that is accessible to a broad audience. 

Paragraph 1 is particularly difficult to parse and needs to be re-evaluated and re-written for clarity, syntax, and grammar.

Paragraph 1 has been corrected.

Paragraph 2 requires references for pannexin channel formation.  Again, the choices for explaining gap junction formation to a general audience are not clear and would require only a small bit of “tweaking” CORRECTED

For example:                                                                                       

Connexins are transmembrane proteins that assemble to form cell-cell channels called gap junctions. Each gap junction is composed of two hemichannels, each of which is made of six connexin subunits. Pannexins are also channel membrane proteins and are formed by six or eight monomers. The way connexins and pannexins form structures within the membrane differ. Connexins, unlike pannexins, provide a mechanism for transport of ions and small signaling molecules between apposed cells. Pannexins have not been shown to form functional cell-cell channels.  Pannexins primarily  form oligomeric structures embedded in a single plasma membrane that when open, provide a conduction pathway between the cytosol and extracellular space.

 Paragraph 3 and 4 also require editing for clarity. CORRECTED

Why choose sheep Cx46 and human PANX1 to test the model?  Why decide to only analyze the hemichannel structure of Cx46?  Pannexins are believed to only form hemichannels, while the hemichannel structure-function of various connexins is a separate area of study from cell-cell channels.

Sheep’s connexin got changed into human.

 RESULTS

A brief explanation of FOD-M and the various analytic features would be helpful.  While a detailed explanation, including derivation of the various elements, is provided in the methods section, acronym definitions in the body of the text are useful for the reader.  This is a manuscript directed towards an educated general audience. To best evaluate the strength of the authors’ conclusions, analysis parameters should be detailed along with presentation of the data.

Fig 1  Figure legend does not provide enough detail identifying the different parts of the structure, including regions of similarity and difference.

CORRECTED – the legends contain the necessary information – it has been added.

The authors conclude that the T, O, and M distributions are typical for membrane proteins as a class but fail to provide references or general data to support this claim.

The different forms of membranÄ™ proteins are discussed in other papers. The comparison with water soluble proteins as well as with amyloids is available in other papers of our group.

Labeling using Pannexin and Connexin (as in Fig 3) would also be helpful. CORRECTED

 Section 2 provides reasonable explanations the “what” and “why” of the analyses presented in the next sections.

 DISCUSSION

The discussion is adequate, but could use some editing for clarity, to adjust awkward phrasing, and fix minor grammatical errors. CORRECTED

Reviewer 3 Report

The manuscript entitled “CONNEXINs and PANNEXINs – similarities and differences according to the FOD-M model” by Roterman et al. applies the FOD-M model to investigate and compare the hydrophobicity profile of a connexin and a pannexin structure. In general, although the manuscript presents a detailed biophysical investigation of the hydrophobicity profile of the two proteins, the physiological consequences of the results are painfully missing from the text. Both results for individual proteins and the comparison of the two profiles should be more deeply interpreted in the context of their physiological function in order to orient the readers of Biomedicines whose major background is in biology and medicine.

Specific remarks:

1)      Several structures of different connexin subtypes have been published in the recent years. What was the rationale of the authors to select Cx46 as their target of investigation? Do the results apply for other subtypes as well? Are there any significant differences in the hydrophobicity profile of different connexin subtypes?

2)      The membrane and extracellular parts of the connexin show much higher K values than their pannexin counterparts. May this difference be attributed to the ability of connexins to form gap junctions (in contrast to pannexins that typically function as single channels)? It would be interesting to get some insight into Cx31.3 (PDB: 6L3T or 6L3U), a connexin that does not form gap junctions.

3)      In many cases, there are significant differences between the RD and especially the K values of the complex and the individual chain. What is the reason for this discrepancy? Since both the connexin and pannexin structures are symmetrical, doesn’t the significant deviation of the complex values come from the variations between the individual chains? If so, what was the rationale to select chain A as a representative chain?

4)      The most interesting conclusion drawn from the analysis is that the extracellular part of the connexin hemichannel shows hydrophobicity profile characteristic to a globular protein – despite it is expected to be docked to the hemichannel from the neighbouring cell. The authors intend to further investigate this observation by analysis of a full gap junction channel. Although it is worth to be done, it should also be noted that the major portion of the extracellular part of connexins is still exposed to water in the full gap junction form and only a small number of residues on each chain are directly involved in docking the two hemichannels. Therefore, analysis of smaller regions of the extracellular parts of a hemichannels would still emerge significant differences in their hydrophobicity profile that may be related to their genuine function.

Minor issues:

Figure 2: the gray line is hardly seen.

The English grammar needs to be significantly improved.

Author Response

Reviewer III

Dear Reviewer

Many thanks for insipring comments. We did our best to follow your advices.

We hope – the current version will be found as satisfactory.

The detailed information given below.

All changed and newly inytroduced frafments are given in red.

Sincerely yours;

Irena Roterman

Yes

Can be improved

Must be improved

Not applicable

Does the introduction provide sufficient background and include all relevant references?

( )

(x)

( )

( )

Are all the cited references relevant to the research?

(x)

( )

( )

( )

Is the research design appropriate?

(x)

( )

( )

( )

Are the methods adequately described?

(x)

( )

( )

( )

Are the results clearly presented?

( )

( )

(x)

( )

Are the conclusions supported by the results?

( )

(x)

( )

( )

Comments and Suggestions for Authors

The manuscript entitled “CONNEXINs and PANNEXINs – similarities and differences according to the FOD-M model” by Roterman et al. applies the FOD-M model to investigate and compare the hydrophobicity profile of a connexin and a pannexin structure. In general, although the manuscript presents a detailed biophysical investigation of the hydrophobicity profile of the two proteins, the physiological consequences of the results are painfully missing from the text. Both results for individual proteins and the comparison of the two profiles should be more deeply interpreted in the context of their physiological function in order to orient the readers of Biomedicines whose major background is in biology and medicine.

Specific remarks:

  • Several structures of different connexin subtypes have been published in the recent years. What was the rationale of the authors to select Cx46 as their target of investigation? Do the results apply for other subtypes as well? Are there any significant differences in the hydrophobicity profile of different connexin subtypes?

The sheep’s pconnexin got substituded by human one.

2)      The membrane and extracellular parts of the connexin show much higher K values than their pannexin counterparts. May this difference be attributed to the ability of connexins to form gap junctions (in contrast to pannexins that typically function as single channels)? It would be interesting to get some insight into Cx31.3 (PDB: 6L3T or 6L3U), a connexin that does not form gap junctions.

This can be the subject of the next analysis. Many thanks for suggestions. We will take it under consideration in our further analysis.

3)      In many cases, there are significant differences between the RD and especially the K values of the complex and the individual chain. What is the reason for this discrepancy? Since both the connexin and pannexin structures are symmetrical, doesn’t the significant deviation of the complex values come from the variations between the individual chains? If so, what was the rationale to select chain A as a representative chain?

 The status expressed for individual chain may be used for interpretation of the chain folding. The question is whether the hydrophobic core is generated ? If so the chain may fold independently without any other intervention than water. The local excess of hydrophobicity on the surface may suggest the potential area for protein complexation.

The status in complex explains the structural role of selected fragments to play the final role of the complex in its biologically active form.

4)      The most interesting conclusion drawn from the analysis is that the extracellular part of the connexin hemichannel shows hydrophobicity profile characteristic to a globular protein – despite it is expected to be docked to the hemichannel from the neighboring cell. The authors intend to further investigate this observation by analysis of a full gap junction channel. Although it is worth to be done, it should also be noted that the major portion of the extracellular part of connexins is still exposed to water in the full gap junction form and only a small number of residues on each chain are directly involved in docking the two hemichannels. Therefore, analysis of smaller regions of the extracellular parts of a hemichannels would still emerge significant differences in their hydrophobicity profile that may be related to their genuine function.

The detailed analysis is taken under consideration for further study.  

Minor issues:

Figure 2: the gray line is hardly seen. CORRECTED

The English grammar needs to be significantly improved.

The Authors are ready to cover the costs of native speaker corrections, if the publisher offers such service.

Round 2

Reviewer 1 Report

I found the reviewed version satisfactory. I listed some comments here:

1. 7F8J was solved using CryoEM. More specifically, they were using nanodisc to stabilize the structure. The author was saying both ref 45&46 were from x-ray which was not correct. One structure using X-Ray and another one using CryoEM raised the question of whether they are comparable. It has been well known due to the way crystals are formed, the membrane structures solved by X-ray might not represent their native form in the native membrane environment. It is worth comparing/mentioning 2ZW3 with other connexins structures solved using CryoEM(especially in nanodisc or liposomes if there are any).

2.."The lengths of the chains are also different: 201 aa (2-109, 125-217) in connexin and 320 aa (2-158, 194-355) in pannexin" .This sentence in the paper was not clear whether the author mutated/truncated the channels or used the whole channel but did not resolve the full channels. I checked both papers and realized they used the whole channel and the missing residues were just not modeled. This sentence should be written like "The lengths of the chains are also different: 226 aa (unresolved regions: 2-109, 125-217) in connexin and 426 aa (unresolved regions:2-158, 194-355) in pannexin.

3. Figure1. Usually, people would plot structures in the same orientation (top view, side view, etc) . Please mention whether C&D are from bottom-up view or top-down view. And change the orientation of E&F to be the same as A&B. Please also change the map contour level of E&F as you can see, F looks much wider than B(a lower threshold). I also found A-D have very poor resolution. I am not sure is it because the high-resolution information was lost in the pdf I am reading or A-D were not saved in a high-resolution setting when the author generated them.

4."The membrane proteins, including the transmembrane ones show the structure adapting to the hydrophobic environment which requires the exposure of hydrophobic residues to the protein surface. " is not very clear. Do you mean "Membrane proteins require the exposure of hydrophobic residues on their surfaces to adapt to the hydrophobic environment."?

5."Currently lipids in form of nanodiscs and liposomes are used to stabilize membrane proteins for both structural and functional analysis [20-21] and lipid nanodisc, liposomes [23]." is not very clear. Do you mean "Currently lipids in form of nanodiscs and liposomes can be used to stabilize membrane proteins for both structural and functional analysis [20-21,23]."?

6."The current status of the membrane proteins recognition is presented in [40]." Here, 40 is a reference number.  Do you want to say "The current progress of the membrane proteins structure researches are presented in some excellent reviews[40 etc]"

7."The special position among the membrane proteins is represented by connexins, pannexins and innexins which differ in structural and functional role playing the common role of channels [40]." This sentence is not clear. It might be better to open with "Connexins, pannexins and innexins ...." instead.

Author Response

REVIEWER I

Dear Reviewer

Many thanks for detailed analysis of our work. The experimental technique may influence the structure characteristics. This is why we substituted 2ZW3 (X-ray) by 6L3t (CryoEM) making both comparised proteins solved by the same experimental technique.

All other comments hav been taken under consideration.

We hope the current version will satisfy your expectations.

Comments and Suggestions for Authors

I found the reviewed version satisfactory. I listed some comments here:

  1. 7F8J was solved using CryoEM. More specifically, they were using nanodisc to stabilize the structure. The author was saying both ref 45&46 were from x-ray which was not correct. One structure using X-Ray and another one using CryoEM raised the question of whether they are comparable. It has been well known due to the way crystals are formed, the membrane structures solved by X-ray might not represent their native form in the native membrane environment. It is worth comparing/mentioning 2ZW3 with other connexins structures solved using CryoEM(especially in nanodisc or liposomes if there are any).

The structure of 2ZW3 got substituted by 6L3T to eliminate the differences in experimental technique applied. Many thanks for directing out attention ofn the experimental technique to make the comparable analysis more reviable. It reveiled the role of M distributions comparison to evaluate the influence of additional environmental compounds – in the discussed case the presence of Ca2+.

2."The lengths of the chains are also different: 201 aa (2-109, 125-217) in connexin and 320 aa (2-158, 194-355) in pannexin" .This sentence in the paper was not clear whether the author mutated/truncated the channels or used the whole channel but did not resolve the full channels. I checked both papers and realized they used the whole channel and the missing residues were just not modeled. This sentence should be written like "The lengths of the chains are also different: 226 aa (unresolved regions: 2-109, 125-217) in connexin and 426 aa (unresolved regions:2-158, 194-355) in pannexin.

The lenghts of chains are different independently on the missing fragments. It got corrected.

  1. Usually, people would plot structures in the same orientation (top view, side view, etc) . Please mention whether C&D are from bottom-up view or top-down view. And change the orientation of E&F to be the same as A&B. Please also change the map contour level of E&F as you can see, F looks much wider than B(a lower threshold). I also found A-D have very poor resolution. I am not sure is it because the high-resolution information was lost in the pdf I am reading or A-D were not saved in a high-resolution setting when the author generated them.

The different orientation was intentional to make possible better visualisation of the complex structures. The resolution is dependent on the source programm delivering the picture (for example the one present in PDB).

4."The membrane proteins, including the transmembrane ones show the structure adapting to the hydrophobic environment which requires the exposure of hydrophobic residues to the protein surface. " is not very clear. Do you mean "Membrane proteins require the exposure of hydrophobic residues on their surfaces to adapt to the hydrophobic environment."?

CHANGED

5."Currently lipids in form of nanodiscs and liposomes are used to stabilize membrane proteins for both structural and functional analysis [20-21] and lipid nanodisc, liposomes [23]." is not very clear. Do you mean "Currently lipids in form of nanodiscs and liposomes can be used to stabilize membrane proteins for both structural and functional analysis [20-21,23]."?

CHANGED

6."The current status of the membrane proteins recognition is presented in [40]." Here, 40 is a reference number.  Do you want to say "The current progress of the membrane proteins structure researches are presented in some excellent reviews[40 etc]"

 CHANGED

7."The special position among the membrane proteins is represented by connexins, pannexins and innexins which differ in structural and functional role playing the common role of channels [40]." This sentence is not clear. It might be better to open with "Connexins, pannexins and innexins ...." instead.

CHANGED

Reviewer 3 Report

In the revised version of their manuscript entitled “CONNEXINs and PANNEXINs – similarities and differences according to the FOD-M model”, Roterman et al. did not appropriately answered my previous concerns. I understand that some of the issues I raised may require additional analysis to be done that the authors may not be in the possession to do. However, at least the other issues need to be properly addressed. Specifically:

1) The physiological consequences of the results are still missing from the text. At least, the authors should describe in the Discussion section what physiologically relevant new knowledge can be deducted from their results.

2) The authors did not answer why they selected Cx46 as their target of investigation. Do the results apply for other subtypes as well? Are there any significant differences in the hydrophobicity profile of different connexin subtypes? Substitution of sheep Cx46 by human Cx46 (as the authors replied) has nothing to do with my concern. This question is especially important as changing even the source of the same protein (sheep vs. human Cx46) significantly modified the K values of the complex (Table 4) and therefore the conclusion drawn by the authors (last paragraph of the Results section). If the results are specific to differences between Cx46 and pannexin-1, this should be clearly stated in the title, the abstract and throughout the text.

3) English grammar and potential misleading sentences should be corrected. For example:

a. Abstract: “Their role is to connect the external environment with the cytoplasm of the cell, where connexin is able to link two cells together allowing the transport from one to another.” should read as “Their common role is to connect the external environment with the cytoplasm of the cell, while connexin is also able to link two cells together allowing the transport from one to another.”

b. Page 2: “Currently lipids in form of nanodiscs and liposomes are used to stabilize membrane proteins for both structural and functional analysis [20-21] and lipid nanodisc, liposomes [23].” should read as “Currently lipids in form of nanodiscs and liposomes are used to stabilize membrane proteins for both structural and functional analysis [20-21]”.

c. Page 2: the sentence “The special position among the membrane proteins is represented by connexins, pannexins and innexins which differ in structural and functional role playing the common role of channels [40].” is difficult to follow.

d. Page 2: “Each gap junction is composed of two hemichannels, which are located in the plasma membrane and adjacent cells.” should read as “Each gap junction is composed of two hemichannels, which are located in the plasma membrane of adjacent cells.”

e.  etc.

Author Response

REVIEWER III

Dear Reviewer

Many thanks for your valuable comments. We did our best to follow your suggestions.

We asked the Editor to take the decission whether additional English correction is necessary.

Sincerely yours;

Irena Roterman

In the revised version of their manuscript entitled “CONNEXINs and PANNEXINs – similarities and differences according to the FOD-M model”, Roterman et al. did not appropriately answered my previous concerns. I understand that some of the issues I raised may require additional analysis to be done that the authors may not be in the possession to do. However, at least the other issues need to be properly addressed. Specifically:

  • The physiological consequences of the results are still missing from the text. At least, the authors should describe in the Discussion section what physiologically relevant new knowledge can be deducted from their results.

The new fragment has been added (Discussion) to present the consequences of modified non-natural external conditions on protein folding – the most sensitive proces depending on external force field. 

  • The authors did not answer why they selected Cx46 as their target of investigation. Do the results apply for other subtypes as well? Are there any significant differences in the hydrophobicity profile of different connexin subtypes? Substitution of sheep Cx46 by human Cx46 (as the authors replied) has nothing to do with my concern. This question is especially important as changing even the source of the same protein (sheep vs. human Cx46) significantly modified the K values of the complex (Table 4) and therefore the conclusion drawn by the authors (last paragraph of the Results section). If the results are specific to differences between Cx46 and pannexin-1, this should be clearly stated in the title, the abstract and throughout the text.

The applicability of fuzzy oil drop model in its modified version (applicability to membranÄ™ conditions) is the object of analysis. So far different membranÄ™ proteins have been analyzed to identify the differences [42-44]. The detailed comparative analysis between the members of particular group and type of membranÄ™ proteins is taken under consideration as the next step of analysis. So far the analysis of K parameter as potential factor for the specificity of folding conditions – external force field – expressing the active participation of environement is in the focus of our research. The detailed analysis of differences between proteins belonging to particular group is planned in the future.

The substitution of sheep Cx46 by human Cx46 was aimed to eliminate the species-dependent characteristics.

The current change of 2ZW3 into 6L3T is due to introducing similar experimental conditions (X-ray and CryoEM) into two proteins examined using common experimental technique which is CryoEM to eliminate the experiment-dependent differences.

3) English grammar and potential misleading sentences should be corrected. For example:

  1. Abstract: “Their role is to connect the external environment with the cytoplasm of the cell, where connexin is able to link two cells together allowing the transport from one to another.” should read as “Their common role is to connect the external environment with the cytoplasm of the cell, while connexin is also able to link two cells together allowing the transport from one to another.”

CORRECTED

  1. Page 2: “Currently lipids in form of nanodiscs and liposomes are used to stabilize membrane proteins for both structural and functional analysis [20-21] and lipid nanodisc, liposomes [23].” should read as “Currently lipids in form of nanodiscs and liposomes are used to stabilize membrane proteins for both structural and functional analysis [20-21]”.

CORRECTED

  1. Page 2: the sentence “The special position among the membrane proteins is represented by connexins, pannexins and innexins which differ in structural and functional role playing the common role of channels [40].” is difficult to follow.

CORRECTED

  1. Page 2: “Each gap junction is composed of two hemichannels, which are located in the plasma membrane and adjacent cells.” should read as “Each gap junction is composed of two hemichannels, which are located in the plasma membrane of adjacent cells.”

CORRECTED
